# The Transcriptional Regulation of Genes Involved in the Immune Innate Response of Keratinocytes Co-Cultured with *Trichophyton rubrum* Reveals Important Roles of Cytokine GM-CSF

**DOI:** 10.3390/jof8111151

**Published:** 2022-10-31

**Authors:** Monise Fazolin Petrucelli, Bruna Aline M. Cantelli, Mozart Marins, Ana Lúcia Fachin

**Affiliations:** 1Biotechnology Unity, University of Ribeirão Preto (UNAERP), Ribeirao Preto 14096-900, Brazil; 2Laboratory of Genetics and Molecular Biology of Fungi, Department of Genetics, Ribeirão Preto Medical School, University of São Paulo, Ribeirao Preto 14049-900, Brazil; 3Medicine Course, University of Ribeirão Preto (UNAERP), Ribeirao Preto 14096-900, Brazil

**Keywords:** dermatophytosis, immune response, epithelial barrier, GM-CSF, cytokines, *Trichophyton rubrum*

## Abstract

*Trichophyton rubrum* is the most causative agent of dermatophytosis worldwide. The keratinocytes are the first line of defense during infection, triggering immunomodulatory responses. Previous dual RNA-seq data showed the upregulation of several human genes involved in immune response and epithelial barrier integrity during the co-culture of HaCat cells with *T. rubrum*. This work evaluates the transcriptional response of this set of genes during the co-culture of HaCat with different stages of *T. rubrum* conidia development and viability. Our results show that the developmental stage of fungal conidia and their viability interfere with the transcriptional regulation of innate immunity genes. The *CSF2* gene encoding the cytokine GM-CSF is the most overexpressed, and we report for the first time that *CSF2* expression is contact and conidial-viability-dependent during infection. In contrast, *CSF2* transcripts and GM-CSF secretion levels were observed when HaCat cells were challenged with bacterial LPS. Furthermore, the secretion of proinflammatory cytokines was dependent on the conidia developmental stage. Thus, we suggest that the viability and developmental stage of fungal conidia interfere with the transcriptional patterns of genes encoding immunomodulatory proteins in human keratinocytes with regard to important roles of GM-CSF during infection.

## 1. Introduction

Cutaneous infections of keratinized tissues, called dermatophytoses, are the most prevalent fungal infections worldwide, affecting about 25% of the world population [1,2]. Although restricted to the cornified layer of the epidermis, dermatophytosis is a chronic opportunistic disease that can be severe in immunocompromised hosts, dramatically decreasing quality of life [3]. The anthropophilic dermatophyte *Trichophyton rubrum* is the most common etiological agent of dermatophytosis worldwide. Infections caused by *T. rubrum* are difficult to treat, and the prevalence of persistent or disseminated infections is high in immunocompromised hosts [4].

During dermatophyte infection, coordinated steps involving adhesion, penetration, and colonization of keratinized tissues are mandatory for the successful establishment of the fungus in the host. An increase in the proliferation of keratinocytes occurs during fungal infection, accompanied by disruption of the epithelial barrier [5]. Thus, keratinocytes are considered the first line of defense, participating directly in the host immune response by recruiting immunomodulatory cells, by secreting signaling molecules called cytokines, and by secreting antimicrobial peptides such as β-defensins and *RNASE7* [6,7,8]. Pathogen recognition is mediated by a range of pattern recognition receptors (PRRs) present on the cell surface that interact with pathogen-associated molecular patterns (PAMP), triggering the innate immune response [9].

The molecular mechanisms that govern host–dermatophyte interactions remain unclear, in particular the transcriptional response of keratinocytes to *T. rubrum* invasion. In a previous work, we evaluated the transcriptional response of a keratinocyte cell line and *T. rubrum* in an infection-like scenario by dual RNA-seq [10]. The data revealed that certain human genes involved in the innate immune response are important for host defense against dermatophytes, including genes encoding antimicrobial peptides, such as *RNASE7*, genes with immunomodulatory effects, such as *SERPINE1* and *SLC11A1*, genes encoding essential components of the epithelial barrier (*FLG* and *KRT1*), and genes encoding cytokines for macrophage recruitment, such as *CSF2*.

Within this scenario, granulocyte-macrophage colony-stimulating factor (GM-CSF) is a cytokine that has been gaining attention due its immunomodulatory effects. In clinical practice, it is used as an adjuvant in vaccine preparations to boost host immunogenicity. GM-CSF is encoded by the *CSF2* gene and stimulates the proliferation and differentiation of granulocytes and macrophages. Preclinical trials have shown that GM-CSF exerts a wide range of immunomodulatory functions in tissues [11]. The clinical use of GM-CSF as an adjuvant has been evaluated for the treatment of cancer such as prostate, skin, breast, and lung cancer in an attempt to boost the antitumor immune response by increasing the efficiency of antigen presentation by dendritic cells and consequently improving T-cell activation [12,13,14]. Prominent levels of GM-CSF have been detected in patients with COVID-19, although its immunomodulatory benefits for a therapeutic approach remain unclear [11,14,15].

Regarding dermatophyte infection, the immunomodulatory role of GM-CSF remains unknown. An antifungal effect of this cytokine has been observed during treatment of oral pseudomembranous candidiasis [16]. Other studies have indicated a therapeutic benefit of GM-CSF in patients with CARD9 (human caspase recruitment domain protein 9) deficiency, which is associated with the spontaneous development of fungal infections that affect the central nervous system [17,18].

Because knowledge of the transcription mechanisms that govern the immune response of human keratinocytes during *T. rubrum*-host interaction is scarce, the present work evaluated the levels expression of several immunomodulatory genes in HaCat keratinocytes during co-culture with different *T. rubrum* fungal elements. In addition, we quantified a set of proinflammatory cytokines.

## 2. Materials and Methods

### 2.1. Strains and Growth Conditions

The *T. rubrum* strain CBS 118892 (CBS-KNAW Fungal Biodiversity Center, Utrecht, The Netherlands) was cultured on Sabouraud dextrose agar for 15 days at 28 °C [19].

### 2.2. Keratinocytes and Growth Conditions

The immortalized human keratinocyte cell line HaCat was purchased from Cell Lines Service GmbH (Eppelheim, Germany). The cells were cultured in RPMI medium (Sigma-Aldrich, St. Louis, MO, USA) supplemented with 10% fetal bovine serum in a humidified atmosphere with 5% CO_2_ at 37 °C. The antibiotics penicillin (100 U/mL) and streptomycin (100 µg/mL) were added to the medium to prevent bacterial contamination [19].

### 2.3. Co-Culture Assay and Conditions

For the co-culture assay, 2 × 10^5^ HaCat cells/mL were plated in 6-well plates and cultured in RPMI medium (Sigma-Aldrich) supplemented with 5% fetal bovine serum at 37 °C in a humidified atmosphere containing 5% CO_2_ for 24 h until cell adherence to the plates.

#### 2.3.1. Conidial Suspension Conditions

A conidial suspension (1 × 10^7^ conidia/mL) was added to HaCat cells, considering the following two phases of conidial development.

Co-culture with conidial phase (COC): the conidial suspension was added directly to the adhered HaCat cells in RPMI medium (Sigma-Aldrich) supplemented with 5% fetal bovine serum in 6-well plates.

Co-culture with pre-germinated conidia (germinative phase, COG): the conidial suspension was added to 5 mL Sabouraud and incubated for 7 h at 28 °C with agitation (130 rpm) to induce the germination of conidia (germ-tube formation). After this period, the conidia were centrifuged to remove the medium and the conidia pellet was resuspended in RPMI medium (Sigma-Aldrich) supplemented with 5% fetal bovine serum and added to adhered keratinocytes plated in 6-well plates.

In both conditions, the co-cultures were re-incubated for 24 and 48 h at 37 °C in an atmosphere containing 5% CO_2_. Uninfected keratinocytes were used as control.

### 2.4. Heat-Inactivated T. rubrum Conidia in the Germinative and Conidial Phase

For the heat-inactivated *T. rubrum* conidia assay, the conidial solutions described in item 2.3.1 were inactivated by heating at 65 °C for 30 min as previously described by [20]. To confirm conidia inactivation, a 100-µL aliquot of the heat-inactivated conidial suspension was plated on Petri dishes and incubated at 28 °C for 7 days to confirm the absence of growth.

After inactivation, the conidial solutions were resuspended in RPMI medium (Sigma-Aldrich) supplemented with 5% fetal bovine serum and added to adhered keratinocytes. Co-cultures with pre-germinated inactivated conidia (COGI) and inactivated conidia (COCI) were incubated at 37 °C in an atmosphere containing 5% CO_2_ for 24 and 48 h.

### 2.5. Human Keratinocyte and T. rubrum Co-Culture Assay with Well Inserts

Keratinocytes were plated as described in item 2.3 in the bottom compartment of a 6-well plate containing the Transwell permeable support (Corning, New York, NY, USA). After cell adherence, a conidial suspension (conidial phase) prepared as described in item 2.3.1 was added to the upper compartment of the well inserts (polycarbonate membrane with a pore size of 0.4 μm) and incubated at 37 °C in an atmosphere containing 5% CO_2_ for 24 and 48 h. This assay was based on the *Candida glabrata* contact-dependent assay described by [21].

### 2.6. Lipopolysaccharide-Keratinocyte Challenge Assay

Lipopolysaccharide (LPS) of *Escherichia coli* was purchased from Sigma-Aldrich and reconstitution was performed according to manufacturer’s instructions. Aliquots of the reconstituted solution were stored at −20 °C until use.

Keratinocytes were plated as described in item 2.3. After cell adherence, the culture medium was replaced with a solution containing RPMI culture medium supplemented with 5% fetal bovine serum without any antibiotics and different concentrations of reconstituted LPS (0.5, 2.5 and 5 µg/mL) [22]. The cells were incubated at 37 °C in an atmosphere containing 5% CO_2_ for 24 and 48 h. Unchallenged keratinocytes were used as control.

### 2.7. Lactate Dehydrogenase Assay

The release of lactate dehydrogenase (LDH) was measured in keratinocyte supernatants of both co-cultures and from the LPS assay to assess cell viability. For the LDH assay, a 50-µL aliquot of cell supernatant of each experimental condition was mixed in a 96-well plate with 100 µL of TOX7 reagent (Sigma-Aldrich) according to the manufacturer’s instructions. Absorbance was read on a Multiscan FC microplate reader (Thermo-Fisher Scientific) at 490 and 690 nm. The percentage of LDH release was calculated as described by [23]. The experiment was performed considering three independent biological replicates according to [10].

### 2.8. Cytokine Secretion during Co-Culture with T. rubrum Fungal Elements and LPS-HaCat Challenge Assay

The proinflammatory cytokines interleukin (IL)-8, IL-6, IL-1β and tumor necrosis factor alpha (TNF-α) were quantified in the supernatant of keratinocytes co-cultured with the germinative or conidial phase of *T. rubrum*. GM-CSF secretion was quantified in the supernatant of keratinocytes co-cultured with the conidial phase of *T. rubrum* and challenged with LPS. Cytokines were quantified in a 96-well plate by enzyme-linked immunosorbent assay (ELISA) using the Quantikine Colorimetric Sandwich assay (PeproTech, Cranbury, NJ, USA) according to manufacturer’s instructions as previously described by [24]. The absorbance of proinflammatory cytokines and GM-CSF was read on a Multiscan FC microplate reader (Thermo-Fisher Scientific) at 620 and 450 nm or 650 and 405 nm, respectively.

### 2.9. RNA Extraction, cDNA Synthesis and RT-qPCR Analysis

Total RNA was extracted using TRI Reagent^®^ (Sigma-Aldrich). RNA concentration and quality were estimated in a MidSci Nanophotometer (Implen, Westlake Village, CA, USA).

For cDNA synthesis, 1 µg of total RNA was initially treated with DNAse 1 Amplification Grade kit (Sigma-Aldrich) to avoid DNA contamination. A High-Capacity cDNA Reverse Transcription^®^ kit (Applied Biosystems, Waltham, MA, USA) was used for cDNA conversion according to the manufacturer’s instructions.

Transcripts were quantified using an Mx3300 q PCR System (Stratagene, San Diego, CA, USA) using the primers listened in Appendix A. Reactions were prepared using a SYBR Taq Ready Mix kit (Sigma-Aldrich) with ROX dye as a fluorescent normalizer according to the manufacturer’s instructions. The cycling conditions were initial denaturation at 94 °C for 10 min, followed by 40 cycles at 94 °C for 2 min, at 60 °C for 60 s, and at 72 °C for 1 min. Gene expression levels were calculated using the 2-ΔΔct comparative method. GAPDH [25] was used as normalizer gene. The results were reported as the mean ± standard deviation of three experiments.

### 2.10. Statistical Analysis

Analysis of variance (ANOVA) was used for statistical analysis. Statistical significance was determined by the Tukey post hoc test considering *p* < 0.05. Data were analyzed using GraphPad Prism 7.0 (GraphPad Software, San Diego, CA, USA).

## 3. Results

### 3.1. Release of Lactate Dehydrogenase by HaCat Keratinocytes during Co-Culture with T. rubrum

Lactate dehydrogenase is a stable cytoplasmic enzyme that is rapidly released in culture supernatants of mammalian cells after rupture of the plasma membrane [26]. Thus, the viability of keratinocytes was evaluated by quantifying LDH release in the cell supernatant in the COC and COG co-culture conditions. The percentage of LDH release ranged from 30–45% after 24 h and from 60–70% after 48 h in the two conditions, respectively.

### 3.2. Fungal Elements of T. rubrum in Different Developmental Stages Promote Distinct Proinflammatory Cytokine Secretion Profiles of Human Keratinocytes

We quantified proinflammatory cytokines (IL-8, IL-6, IL-1β, and TNF-α) to evaluate the response of keratinocytes during co-culture with *T. rubrum*. The results are shown in Figure 1. The concentrations (pg/mL) of IL-8, IL-6 and TNF-α secreted by keratinocytes differed significantly according to co-culture condition (COC and COG).

All cytokines were secreted into the cell supernatant in at least one of the co-culture conditions evaluated. In both conditions (COC and COG), higher secretion of IL-8 was observed only at 48 h. The concentrations of IL-6 were the same at 24 h in both conditions, while a reduction in the concentration of this cytokine was observed at 48 h in the COC condition.

TNF-α levels were elevated only in the COG condition at 24 h and in the COC condition at 48 h. On the other hand, secretion of IL-1β was slightly higher at 24 h in the COG condition, while higher levels were observed at 48 h in both co-culture conditions.

### 3.3. Analysis of Expression of Human Genes Involved in the Innate Immune Response and Epithelial Barrier Integrity during Co-Culture with T. rubrum

To gain further insight into the modulation of the genes identified in the dual-RNAseq experiment previously published by our research group [10], we evaluated the expression levels of the human genes *CSF2*, *RNASE7*, *SERPINE1*, *SLC11A1*, *KRT1*, and *FLG* in the COC and COG co-culture conditions at 24 and 48 h (Figure 2). Among the genes involved in the host’s innate immune response, the *CSF2* gene was found to be upregulated (higher fold change), especially in the COC condition, with higher expression at 48 h. Co-culture in the COC and COG conditions promoted differences in the expression levels of the *SERPINE1*, *FLG*, and *SLC11A1* genes, suggesting that the expression of genes involved in the immune response of keratinocytes is affected by the stage of fungal development (conidia or germinative phase).

### 3.4. Analysis of Expression of Genes Involved in the Innate Immune Response of Keratinocytes Co-Cultured with Heat-Inactivated Fungal Elements of T. rubrum

Expression analysis of human genes involved in the immune response, and epithelial barrier integrity of keratinocytes using heat-inactivated fungal elements in different developmental stages was used as a strategy to assess whether the modulation of expression of these genes differed between live and dead *T. rubrum* fungal elements. The results are shown in Figure 3. Lower modulation, especially of the *CSF2* gene, was observed when heat-inactivated fungal elements were used. Lower expression of other genes such as *SLC11A1*, *SERPINE1*, and *RNASE7* was found in the COCI and COGI culture conditions as well. Interestingly, the epithelial barrier gene *FLG* showed the opposite modulation profile in response to the inactivation of *T. rubrum* fungal elements at 24 h compared to the other genes described. This gene was induced within 24 h in the COCI and COGI conditions, while the gene was repressed in the COC and COG conditions (Figure 2). In addition, discrete induction of the *KRT1* gene was observed in the COCI and COGI co-culture conditions.

Gene expression analysis in the different co-culture conditions (COC, COG, COCI, and COGI) showed that, among the genes involved in the innate immune response of keratinocytes previously screened by dual RNA-seq, the *CSF2* gene exhibited the highest levels of induction. The results of the present study help to clarify several important aspects of the role of this gene in the response of keratinocytes to *T. rubrum*.

### 3.5. Analysis of CSF2 Expression during Co-Culture of Keratinocytes and T. rubrum Separated by Permeable Inserts

To evaluate the need for cell contact between human keratinocytes and *T. rubrum* for the induction of *CSF2* expression, we carried out a co-culture assay in the COC condition (the condition in which the *CSF2* was most expressed, Figure 2) using permeable inserts that prevented contact with and adherence of *T. rubrum* to keratinocytes. The results shown in Figure 4 indicate that the induction of *CSF2* is dependent on the contact with and adherence of *T. rubrum* to keratinocytes at 24 and 48 h of co-culture.

### 3.6. CSF2 Gene Expression in Keratinocytes Challenged with Bacterial LPS

The expression levels of the *CSF2* gene were evaluated in human keratinocytes challenged with different concentrations of *E. coli* LPS. Within this context, the viability of keratinocytes challenged with 0.5, 2.5, and 5 µg/mL of LPS for 24 and 48 h was assessed by the LDH assay. Release of this enzyme higher than 40% was only observed at the concentration of 5 µg/mL LPS after 48 h when compared to control.

The regulation of Toll-like receptor 4 (TLR4) mRNA was analyzed to determine whether the LPS concentrations used were adequate to challenge HaCat cells. This gene is involved in the cell recognition of bacterial LPS. All LPS concentrations tested induced the expression of TLR4 mRNA (data not shown).

The results of *CSF2* mRNA expression analysis showed induction of this gene in keratinocytes challenged with 2.5 and 5 µg/ mL LPS at 24 h and challenged with all LPS concentrations tested at 48 h (Figure 5). However, different levels of *CSF2* expression were observed when cells co-cultured with *T. rubrum* and challenged with LPS were compared. In co-culture of keratinocytes with *T. rubrum*, the fold change values of *CSF2* mRNA ranged from 10–30 (Figure 4), while in keratinocytes challenged with LPS the fold change ranged from 1.5–3 (Figure 5).

### 3.7. Secretion of GM-CSF by Keratinocytes Co-Cultured with T. rubrum and Challenged with Bacterial LPS

Cytokine GM-CSF, the product of the *CSF2* gene, was quantified in the supernatants of keratinocytes co-cultured in the COC condition (the condition in which the *CSF2* gene was more expressed, Figure 2) and of keratinocytes challenged with different concentrations of LPS.

The results showed that GM-CSF was only secreted when keratinocytes were co-cultured with *T. rubrum* (Figure 6). However, when the cells were challenged with LPS, no secretion of GM-CSF was detected at concentrations above the minimum detection limit of the kit used (32 pg/mL).

## 4. Discussion

Dermatophytoses are one of the most common fungal infections worldwide, and *T. rubrum* is the most important clinical species, causing approximately 69.5% of all dermatophytoses in humans [4,27]. Despite its clinical importance, little is known about the mechanisms that govern the immune response of human keratinocytes in this disease, especially at the molecular level.

Our group evaluated for the first time the transcriptional profile of human HaCat keratinocytes co-cultured with pre-germinated *T. rubrum* conidia [10] and identified genes that are important for the innate immune response of keratinocytes. Considering that the lifecycle of dermatophytes comprises two important growth phases (conidial and mycelial) [28], the present study evaluated cellular features such as the secretion of proinflammatory cytokines, and compared the modulation of expression of genes involved in the human immune response during co-culture of keratinocytes with two different types of fungal elements of *T. rubrum*, namely, conidial and germinative.

Regardless of the developmental stage of the fungus used for co-culture (COC or COG), the release of LDH by human keratinocytes reached 30–45% within 24 h, a percentage similar to that reported in previous studies [10,29]. After 48 h of co-culture, the percentage of LDH release was 60–70% for COC and COG, respectively. The greater LDH release at 48 h might be related to the development of fungal structures adhered to keratinocytes (data not shown).

### 4.1. The Levels of Proinflammatory Cytokine Secretion by Human Keratinocytes May Be Related to the Stage of Fungal Development

Proinflammatory cytokines such as IL-8, IL-6, IL-1β and TNF-α were secreted into the cell supernatant during co-culture of HaCat cells with different developmental stages of *T. rubrum* (COC and COG). At 24 h of co-culture, higher levels of IL-1β and TNF-α were observed in the COG condition and higher levels of IL-6 in the COG and COC conditions when compared to control (uninfected cells). In the COG developmental stage, lower levels of IL-8 were detected in co-cultured cells compared to control. At 48 h, higher levels of IL-8 and IL-1β were observed in both co-culture conditions and higher levels of TNF-α in the COC condition. In the same condition, lower levels of IL-6 were detected in co-cultured cells compared to control (Figure 1).

Keratinocytes are the most abundant cells in the epidermis, and are key players in the initial immune response to dermatophytes and their antigens. These cells actively participate in defense against these infections by recruiting or activating other cells of the immune system through the release of multiple inflammatory cytokines, including the potent chemotactic factor of neutrophils IL-8, IL-6 and TNF-α, which together exert important activity in the tissue inflammatory response [30].

Anthropophilic dermatophytes such as *Trichophyton* species are known to induce the production of cytokines such as IL-8, IL-6, IL-1β, and TNF-α [31,32,33]. However, our results suggest that the stage of fungal growth can influence the secretion profile of these cytokines. Analysis of the effect of *Trichophyton* species such as *T. mentagrophytes*, *T. tonsurans*, and *T. rubrum* on cytokine production by normal human epidermal keratinocytes showed higher levels of IL-8, GRO-α, TNF-α, and GM-CSF after co-culture for 3–24 h. In addition, differences in the production level of each cytokine were observed according to dermatophyte species [33]. Furthermore, higher levels of IL-8 and TNF-α were demonstrated in the supernatant of human epidermal keratinocytes derived from donors and co-cultured with *T. mentagrophytes* [32].

### 4.2. The CSF2 Gene May Play an Important Role in the Immune Response of Keratinocytes

Dual RNA-seq data of human keratinocytes and *T. rubrum* showed the modulation of genes that are important for host immune defense, including genes grouped into functional categories involved in the MAPK pathway, antimicrobial immune response, and establishment of the epithelial barrier [10]. Because the molecular mechanisms involved in the host response to dermatophytes are poorly understood, we selected a number of genes (*CSF2*, *SERPINE1*, *SLC11A1*, *RNASE7*, *FLG* and *KRT1*) in order to evaluate their modulation in COC and COG co-culture conditions and after heat inactivation of *T. rubrum* fungal elements (COCI and COGI).

The gene encoding ribonuclease 7 (*RNASE7*) was induced after both periods, and there was no significant difference between the COC and COG conditions (Figure 2). The release of antimicrobial peptides such as ribonuclease 7 is important for the initial defense of the epidermal layer against fungal infections [6,7,34], and these peptides can even inhibit the growth of dermatophytes such as *T. mentagrophytes*, *Microsporum canis*, *T. rubrum*, and *Epidermophyton floccosum* [35]. Furthermore, the *SLC11A1* gene (Nramp1), which is known to increase host protection against intracellular pathogens [36], was upregulated at 48 h when co-cultured in the COC condition.

Another gene that showed promising results was *SERPINE1*, which was induced within 24 h in the COC condition (Figure 2). No direct relationship between the participation of this gene and the human immune response to dermatophytes has been reported, although in silico analyses have indicated *SERPINE1* as an inducer of the immune response to the pathogenic fungi *Aspergillus fumigatus* and *Candida albicans* [37]. Furthermore, the initial host response to skin and soft tissue infections caused by the Gram-positive bacterium *Staphylococcus aureus* seems to be dependent on the expression of *SERPINE1* [38]. Because *SERPINE1* showed greater induction after 24 h of co-culture in the COC condition, we suggest that this gene may participate in the initial immune response of keratinocytes to *T. rubrum*.

Regarding the genes involved in the maintenance of the epidermal barrier, the gene encoding filaggrin (*FLG*) was found to be repressed both at 24 and 48 h in the COG condition. In the COC condition, this gene was repressed only at 24 h, suggesting differences in the expression profile of this gene according to the stage of fungal development. The gene encoding keratin 1 (*KRT1*) was downregulated at each of the two time points, and there were no significant differences between the COC and COG conditions. These data suggest destabilization of genes that encode key proteins for the composition and integrity of the epidermal barrier. Failures in the composition of these components can facilitate the spread of the fungus in host tissue [39].

Considering all genes evaluated in this study, *CSF2*, which encodes cytokine GM-CSF, was the most upregulated gene, especially after 48 h of co-culture. Comparison of the COC and COG conditions showed more expressive induction of this gene in the former condition (Figure 2), suggesting differences in the expression pattern of this gene according to the developmental stage of conidia.

Although the role of GM-CSF in dermatophytosis remains unknown, an increase in the secretion of this cytokine was observed during co-culture of keratinocytes with the dermatophyte *Trichophyton benhamiae* [12]. Furthermore, studies on *Candida glabrata* have shown that GM-CSF is the main cytokine induced during interaction with human oral keratinocytes. Topical application of this cytokine as a mouthwash in patients with oral pseudomembranous candidiasis was effective in reducing ulcerative lesions, favoring neovascularization and tissue repair of the mucosa [16,21].

Notwithstanding this, during fungal invasion the keratinocytes cells increase their proliferation, recruitment, and signaling for immune cells towards the site of infection [40]. In this sense, the upregulation of *CSF2* and GM-CSF levels could suggest a strategy for stimulating the proliferation and differentiation of granulocytes and macrophages for host defense. Reports have shown that under certain kinds of pathogen attacks (e.g., bacterial, fungal, and virus infections), the content of nitrogen-containing polyamine compounds increases drastically, leading to the triggering of macrophage apoptosis and affecting the activity of transcription factors such as NF-kb and AP-1 that govern important roles for immune response [41]. On the other hand, M2 macrophages exert regulatory effects, increasing polyamines to stimulate cell growth and repair. [42,43]. Further studies are needed to clarify the role of polyamines in dermatophyte infections; however, we hypothesize that the upregulation of *CSF2* could exert an immunoregulatory role during fungi–host interaction, as its modulation is required for macrophage recruitment that is needed during interplay with increased polyamines in the epidermis for tissue repair.

To our knowledge, there are no studies evaluating the antifungal effect or role of GM-CSF in infections with the dermatophyte *T. rubrum*. Thus, our results provide new insights into the modulation profile of *CSF2* and GM-CSF secretion by human keratinocytes in an infection-like scenario using the main global causative agent of dermatophytosis.

### 4.3. The CSF2 Gene of Keratinocytes Is Less Expressed during Co-Culture with Heat-Inactivated T. rubrum Fungal Elements

We evaluated the expression of the *CSF2*, *SERPINE1*, *SLC11A1*, *RNASE7*, *KRT1*, and *FLG* genes during co-culture of human keratinocytes with *T. rubrum* using heat-inactivated fungal elements (COCI and COGI). Even inactivated fungal elements contain cell wall proteins that can stimulate the immune response [44,45].

Our results showed that inactivated fungal elements promoted the downregulation of genes directly involved in the immune response of human keratinocytes (*SERPINE1*, *SLC11A1*, *RNASE7*), particularly of the *CSF2* gene, which exhibited a greater reduction in fold change values compared to its expression profile in the presence of live fungal elements (Figure 3). These results suggest that the use of inactivated *T. rubrum* fungal elements stimulates a lower immune response in HaCat cells by the genes evaluated here.

Regarding the genes involved in epithelial barrier integrity (*FLG* and *KRT1*), interestingly, the *FLG* gene was induced within 24 h in the COCI and COGI conditions (Figure 3), opposite to the profile observed when the conidia were viable (Figure 2). The *KRT1* gene remained repressed in both conditions. The downregulation of the *FLG* gene in the presence of live conidia suggests that *T. rubrum* destabilizes the epithelial barrier by repressing the *FLG* gene. On the other hand, in the presence of inactivated fungal elements, this gene starts to be upregulated. The heat-inactivated cell wall of *T. rubrum* can undergo remodeling that induces expression of the *FLG* gene.

These results show that the use of heat-inactivated *T. rubrum* conidia significantly reduces the expression of PRRs in HaCat keratinocytes, suggesting that, when inactivated, the cell wall antigens of this fungus elicit a lower immune response in these cells, consequently limiting cytokine secretion [46].

Taken together, these data indicate important points for a better understanding of how certain human genes involved in the immune response to *T. rubrum* can be modulated when they are exposed to different fungal elements, either in the conidial phase with previous germination or inactivated by heat. Furthermore, we showed that the secretion profile of certain inflammatory cytokines is influenced by the tested conditions.

We observed significant induction of the *CSF2* gene during co-culture of HaCat keratinocytes with viable fungal elements of *T. rubrum*. Therefore, this gene and its product (GM-CSF) became the focus of this study in order to further analyze the behavior of this cytokine in fungus–host interaction.

### 4.4. The Expression of CSF2 Is Dependent on the Contact between Human Keratinocytes and T. rubrum Fungal Elements

Our results showed that the expression of *CSF2*, and consequently the secretion of GM-CSF, is dependent on the contact between HaCat keratinocytes and *T. rubrum* (Figure 4). This is the first study to establish this contact-dependent relationship between *T. rubrum* and keratinocytes for *CSF2* expression or GM-CSF secretion. Similarly, [21] observed that GM-CSF secretion by oral epithelial cells is dependent on the adhesion of host cells to *Candida glabrata*.

### 4.5. Expression of the CSF2 Gene Was Lower When HaCat Keratinocytes Were Challenged with Bacterial LPS

Another important aspect of the *CSF2* gene was to evaluate its modulation profile when HaCat keratinocytes were challenged with different concentrations of LPS. For this purpose, keratinocytes were challenged with three different concentrations of LPS (0.5, 2.5 and 5 µg/mL), which were sufficient to induce the gene encoding TLR4 of the PRR family. This receptor recognizes LPS present on the cell wall of Gram-negative bacteria, and is expressed on the surface of hematopoietic cells, monocytes, dendritic cells, macrophages, and human epidermis [47,48].

Our results showed differential levels of *CSF2* expression when HaCat keratinocytes were challenged with LPS or co-cultured with *T. rubrum*. When HaCat keratinocytes were challenged with various concentrations of LPS, *CSF2* was upregulated in 24 h and 48 h post-challenge. However, the fold changes ranged from 5–30 during co-culture with *T. rubrum* (Figure 2), while challenging keratinocytes with LPS resulted in fold changes of CSF ranging from 1–3 (Figure 5), suggesting that this gene may be differently regulated when keratinocytes are challenged with LPS of gram-negative bacteria and in the presence of *T. rubrum*.

### 4.6. The Secretion of GM-CSF by Human Keratinocytes Was Greater at 48 h during Co-Culture with T. rubrum, and Was Not Detected during Co-Culture with LPS

After expression analyses demonstrated strong induction of the *CSF2* gene when HaCat keratinocytes were co-cultured in the COC condition, we evaluated the secretion of GM-CSF in the cell culture supernatant. The highest concentration of GM-CSF in the supernatant was reached at 48 h (Figure 6), corroborating the gene expression data, in which the time interval of highest *CSF2* induction was 48 h as well (Figure 2). GM-CSF secretion was not detected when keratinocytes were challenged with LPS.

During the inflammatory stage, GM-CSF is particularly important for wound healing, and in vitro studies have shown that this cytokine increases keratinocyte proliferation, promoting re-epithelialization and restructuring of the epithelial barrier [49]. The role of GM-CSF in dermatophytoses caused by *T. rubrum* remains unclear; however, we suggest that this cytokine is strongly induced, especially at 48 h, to help with epithelial barrier restructuring. Epithelial damage is known to occur during fungus–host interaction, which is triggered by penetration of the fungus and secretion of keratinolytic proteases.

Recent studies have indicated the use of GM-CSF as an adjuvant in vaccine preparations to increase immunogenicity and to elicit a more efficient immune response. GM-CSF has already been tested in animal and human studies for anti-tumor immunotherapy in prostate, skin, breast, and lung cancer with divergent results [14,50]. Moreover, recombinant GM-CSF, marketed as Sargramostim (Sanofi, Paris, France), has been approved for clinical studies, including the treatment of neutropenia in bone marrow transplant recipients, patients undergoing chemotherapy, and carriers of human immunodeficiency virus [51].

With respect to antifungal therapy, the limited availability of highly selective antifungals, along with the emergence of strains resistant to the drugs currently in use, highlights the need to develop new therapeutic strategies in order to obtain more promising and beneficial treatment results for patients. One such strategy is to increase host immunity, for example, through immunotherapy using cytokines [52]. Within this scenario, GM-CSF has shown promising results as an immunological adjuvant for the treatment of ventriculitis caused by *Aspergillus* species, in which it was combined with voriconazole, amphotericin B and caspofungin [52]. Infection with *Scedosporium apiospermum* was successfully treated with the combination of cafungin and GM-CSF [53]. Therapeutic improvement was observed when this cytokine was used for the treatment of pseudomembranous candidiasis caused by *C. glabrata* [54].

It is too early to say whether GM-CSF therapy elicits a better immune response to infections caused by *T. rubrum*, either alone or in combination with commercial antifungals. However, for the first time our results show strong induction of the *CSF2* gene and the consequent secretion of GM-CSF during the interaction of *T. rubrum* with HaCat keratinocytes. Furthermore, we demonstrated that contact between fungal and host cells is required, and that fungal viability is necessary for gene and cytokine expression during co-culture, which simulates a superficial infection.

## Figures and Tables

**Figure 1 jof-08-01151-f001:**
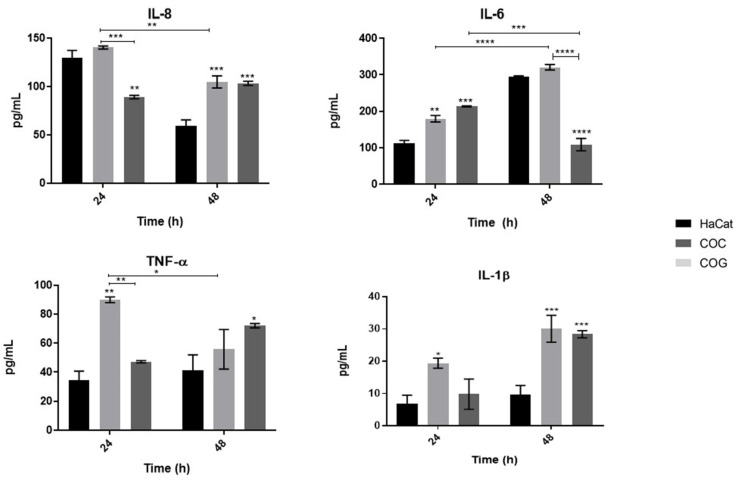
Secretion of cytokines by human keratinocytes during co-culture with fungal elements of *T. rubrum*. HaCat: cultured keratinocytes used as control; COC: co-culture with the conidial phase of *T. rubrum*; COG: co-culture with the germinative phase of *T. rubrum*. Asterisks indicate significant differences (* *p* < 0.05; ** *p* < 0.01; *** *p* < 0.001; **** *p* < 0.0001).

**Figure 2 jof-08-01151-f002:**
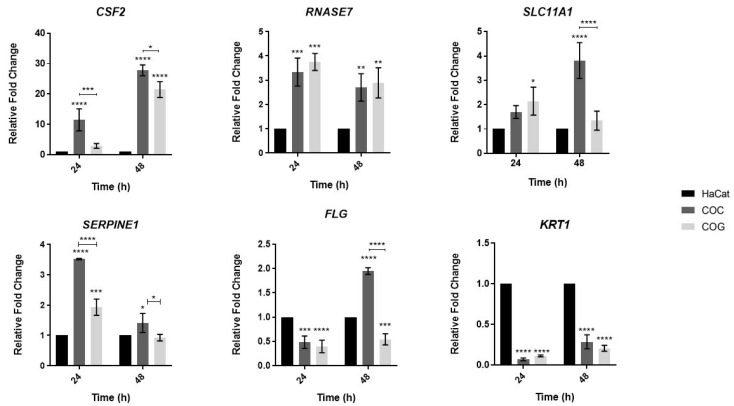
Expression levels of genes involved in the innate immunity (*CSF2*, *RNASE7*, *SLC11A1*, and *SERPINE1*) and epithelial barrier integrity (*KRT1* and *FLG*) of human keratinocytes in response to fungal elements of *T. rubrum*. HaCat: cultured keratinocytes used as control; COC: co-culture with the conidial phase of *T. rubrum*; COG: co-culture with the germinative phase of *T. rubrum*. Asterisks indicate significant differences (* *p* < 0.05; ** *p* < 0.01; *** *p* < 0.001; **** *p* < 0.0001).

**Figure 3 jof-08-01151-f003:**
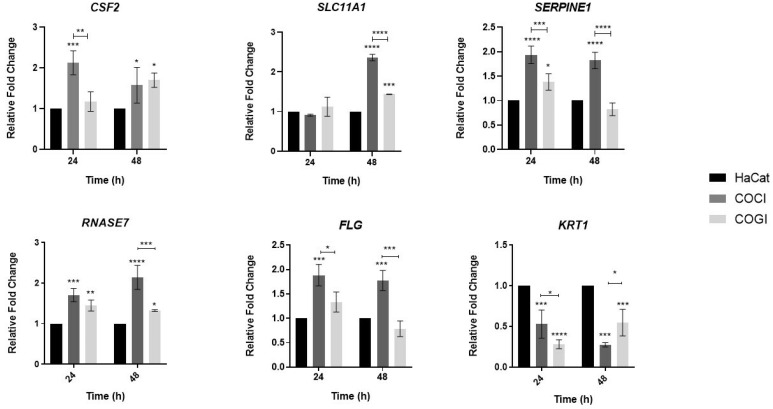
Expression levels of genes involved in the innate immunity (*CSF2*, *RNASE7*, *SLC11A1*, and *SERPINE1*) and epithelial barrier integrity (*KRT1* and *FLG*) of human keratinocytes in response to heat-inactivated fungal elements of *T. rubrum*. HaCat: cultured keratinocytes used as control; COCI: co-culture with heat-inactivated conidial phase of *T. rubrum*; COGI: co-culture with heat-inactivated germinative phase of *T. rubrum*. Asterisks indicate significant differences (* *p* < 0.05; ** *p* < 0.01; *** *p* < 0.001; **** *p* < 0.0001).

**Figure 4 jof-08-01151-f004:**
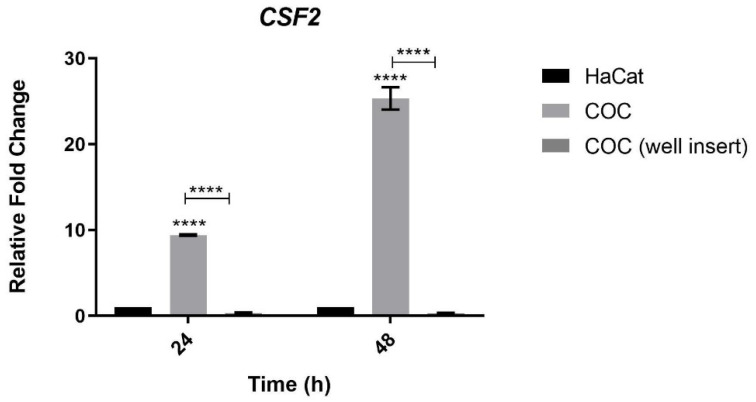
*CSF2* gene expression levels during co-culture of human keratinocytes and *T. rubrum* with well inserts. HaCat: cultured keratinocytes used as control; COC: co-culture with the conidial phase of *T. rubrum*; COC (well insert): co-culture with the conidial phase of *T. rubrum* separated by well inserts (without cell–cell contact between the pathogen and host). Asterisks indicate significant differences (**** *p* < 0.0001).

**Figure 5 jof-08-01151-f005:**
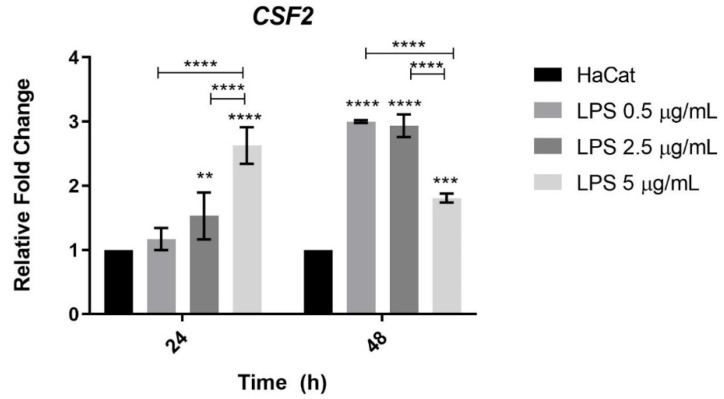
*CSF2* gene expression levels in human keratinocytes challenged with LPS. HaCat: keratinocytes were used as control. Three different concentrations of LPS were tested (0.5, 2.5, and 5 µg/ mL). Asterisks indicate significant differences (** *p* < 0.01; *** *p* < 0.001; **** *p* < 0.0001).

**Figure 6 jof-08-01151-f006:**
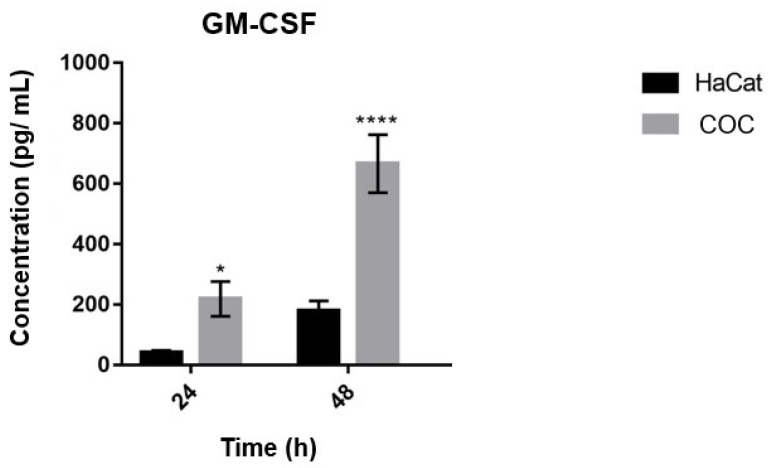
Quantification of GM-CSF cytokine during co-culture of human keratinocytes and *T. rubrum*. Cultured keratinocytes as used as control (HaCat). COC indicates co-culture with conidial phase of *T. rubrum*. Significantly different values are shown by asterisks, and were determined using ANOVA followed by Turkey’s post hoc test (* *p* < 0.05; **** *p* < 0.0001).

## Data Availability

Not applicable.

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
