# Peer review of "The Transcriptional Regulation of Genes Involved in the Immune Innate Response of Keratinocytes Co-Cultured with *Trichophyton rubrum* Reveals Important Roles of Cytokine GM-CSF"

_jof, 2022, doi:10.3390/jof8111151_

Round 1
Reviewer 1 Report
The authors present co-culture experiments of human keratinocytes with Trichophyton rubrum conidia of two growth stages. The effects of secreted cytokines of keratinocytes were measured and transcriptional levels of genes involved in the innate immunity analyzed.
Viability of fungal cells were important to induce keratinocyte response especially of CSF2 expression.
Minor points:
Lines 95, 96: “A conidial suspension….” What is the relation between HaCat cells and fungal conidia? How many mL (or µL) of conidia suspension were added to how many volume of HaCat cell concentration described in line 91?
Line 99-100: “the conidial suspension…” Do the authors control the efficiency of germination under the chosen conditions? How many percent of conidia show germ tubes after pretreatment?
Line 326: “Arthroderma benhamiae” The new taxonomical system of deHoog et al. 2017 (Mycopathol 182:5-31) integrates teleomorph and anamorph species names in one system. The accepted name today is Trichophyton benhamiae.
References:
The authors should check the journal style. Most species names do not written in italic letters.
Author Response
Reviewer Answers
Reviewer nº1
- Lines 95, 96: “A conidial suspension….” What is the relation between HaCat cells and fungal conidia? How many mL (or µL) of conidia suspension were added to how many volume of HaCat cell concentration described in line 91?
Author’s answer: The keratinocyte cells were counted in a hemocytometer for the co-culture assay and adjusted for 2 x 105 cells/mL. We used 3 mL for each well (6-well plate). So, we have 6 x 105 cells/ well in the final volume. After 24 h of keratinocyte cell adherence, we add a conidia suspension adjusted for 1 x 107 conidia/mL. Thus, the cultured medium was replaced by 3 mL of RPMI medium with conidia suspension. We have a total of 3 x 107 conidia/well in the final volume.
- Line 99-100: “the conidial suspension…” Do the authors control the efficiency of germination under the chosen conditions? How many percent of conidia show germ tubes after pretreatment?
Author’s answer: In a previous work (unpublished data), our group monitored the conidia germination in Sabouraud liquid medium for 24 h. We observed that in 7 hours occurs, the germ tube formation in most of the conidia (please see the figure above). Thus, we used this time point for a pre-germination treatment before fungi inoculation in keratinocytes cells.
- rubrum conidia germination in Sabouraud medium at 28°C at different time points. The arrows indicate the stage of the conidia development. In 3 h, we observed only conidia. In 7 h, there is the initial germ tube formation. In 14 h, we observed the hyphae development, and in 24 h, the mycelium formation.
- Line 326: “Arthroderma benhamiae” The new taxonomical system of deHoog et al. 2017 (Mycopathol 182:5-31) integrates teleomorph and anamorph species names in one system. The accepted name today is Trichophyton benhamiae.
Author’s answer: Thank you for your suggestions. The name was changed to Trichophyton benhamiae. Please, see the new version of the manuscript.
- The authors should check the journal style. Most species names do not written in italic letters.
Author’s answer: Thank you for your suggestion. The references were revised in the author’s guidelines and performed with a Reference Manager with the Journal of Fungi style that uses italic letters only for the Journal names.

Reviewer 2 Report
The authors represent a novel report on interplay between human keratinocytes and Trichophyton rubrum-mediated fungal infection. Extensive trascriptional analysis performed in this study allowed the authors to identify several genes involved in the cellular innate immune response and in the maintenance of epithelial barrier integrity during the infection. Excitingly, CSF2 encoding the cytokine GM-CSF was the most upregulated gene – this is a first report that CSF2 expression during T. rubrum-host interaction is contact-dependent. The study deserves attention and demonstrated a good level of experimental validation and presentation. Methods used are convincing (RT-qPCR, RNAseq, etc.) for this kind of study.
Some suggestions for improvement:
Line 89 – a dot is missing after [19] → [19].
Line 191, 211, 223, 242 – please review the error bars for HaCat (black column). While in the Fig. 1 and 6 error bars for this sample are visible, in other figures they seem to be missing.
Line 292 – I suggest to shift the citation ([32]) mention to the end of the sentence for better reading.
Line 302-409 – please review the figure mentions, sometimes figure references are in bold (like in the results part), sometimes they are not in bold
Lines 321-333 – I suggest to discuss this interesting observation in some more detail.
It was reported that in human skin at site of infection content of nitrogen-containing compounds polyamines increase dramatically in both proliferating cells and extracellular tissue fluids during inflammation, tissue regeneration and cell damage. Increased polyamines levels have been demonstrated at inflammatory sites of bacterial, fungal, viral infections (doi: 10.1074/jbc.M611686200). On the other hand, a metabolic shift in M2-macrophages allows their recruitment at the site of infection and damaged tissue to repair/heal the tissue (doi: 10.3389/fimmu.2014.00603). May the upregulation of the CSF2 gene during the T. rubrum-host interaction be required for stimulation of the proliferation and differentiation of granulocytes and macrophages, which are then needed in interplay with enchanced polyamine production in epidermis for repair of damaged or infected skin tissue/keratinocytes?
Line 461 – the names of authors in this citation are all abbreviated, the surnames should be readable
Line 545 – please review the names in this citation

Author Response
Reviewer nº2
- Line 89– a dot is missing after [19] → [19].
Author’s answer: Thank you for your suggestions. This correction was performed in the new version of the manuscript.
- Line 191, 211, 223, 242 – please review the error bars for HaCat (black column). While in the Fig. 1 and 6 error bars for this sample are visible, in other figures they seem to be missing.
Author’s answer: Figures 1 and 6 correspond to a cytokine secretion graph. Thus, there are error bars in Hacat (control). Other figures corresponded to RT-qPCR analyses, and in this case, we do not use the error bar for HaCat because it’s considered a calibrator reference. As we performed a relative expression analysis, the HaCat values have a baseline value of 1 for comparison between co-culture conditions. Thus, values of relative expression above 1 are considered an upregulation, and below 1 are considered a downregulation.
- Line 292 – I suggest to shift the citation ([32]) mention to the end of the sentence for better reading.
Author’s answer: Thank you for your suggestions. This correction was performed in the new version of the manuscript.
- Line 302-409 – please review the figure mentions, sometimes figure references are in bold (like in the results part), sometimes they are not in bold
Author’s answer: Thank you for your suggestions. This correction was performed in the new version of the manuscript.
- Lines 321-333 – I suggest to discuss this interesting observation in some more detail. It was reported that in human skin at site of infection content of nitrogen-containing compounds polyamines increase dramatically in both proliferating cells and extracellular tissue fluids during inflammation, tissue regeneration and cell damage. Increased polyamines levels have been demonstrated at inflammatory sites of bacterial, fungal, viral infections (doi: 10.1074/jbc.M611686200). On the other hand, a metabolic shift in M2-macrophages allows their recruitment at the site of infection and damaged tissue to repair/heal the tissue (doi: 10.3389/fimmu.2014.00603). May the upregulation of the CSF2 gene during the rubrum-host interaction be required for stimulation of the proliferation and differentiation of granulocytes and macrophages, which are then needed in interplay with enchanced polyamine production in epidermis for repair of damaged or infected skin tissue/keratinocytes?
Author’s answer: Thank you for your suggestions. We included this exciting point in the manuscript (please see lines 330-340 of the new manuscript). In fact, during a fungi-host interaction, keratinocyte cells increase their proliferation possibly due to increased polyamine compounds. Thus, M2 macrophages can also stimulate polyamines to stimulate cell growth and repair. So, we hypothesize that csf2-mRNA and GM-CSF could exert immunoregulatory roles in the recruitment of macrophages to the site of infection that is needed during an interplay with an increased polyamine in the epidermis for tissue repair.
- Line 461 – the names of authors in this citation are all abbreviated, the surnames should be readable
Author’s answer: Thank you for your suggestions. This correction was performed in the new version of the manuscript.
- Line 545 – please review the names in this citation
Author’s answer: Thank you for your suggestions. This correction was performed in the new version of the manuscript.

Reviewer 3 Report
The authors Petrucelli and colleagues of the original research entitled “The transcriptional regulation of genes involved in the immune innate response of keratinocytes co-cultures with Trichophyton rubrum reveals important roles of cytokines GM-CSF” aimed to evaluate the levels of expression of several immunomodulatory genes in HaCat keratinocytes co-cultures with Trichophyton rubrum. Globally, the work is very updated and of general interest to the journal readers. The methods used are diversified and the results obtained are straightforward and interesting.
However, some minor issues should be addressed before acceptance.
Minor comments
- The abstract should be shortened
- Line 42 – please replace superficial with cutaneous
- Line 79-80 – the last sentence of the introduction is a result. Please delete it.
Author Response
Reviewer nº3
- The abstract should be shortened
Author’s answer: Thank you for your suggestions. The abstracted was restructured. Please see in the new version.
- Line 42 – please replace superficial with cutaneous
Author’s answer: Thank you for your suggestion. We replaced the word.
- Line 79-80 – the last sentence of the introduction is a result. Please delete it.
Author’s answer: Thank you for your suggestion. We delete the sentence.
